# COVID-19 Vaccination and Serological Profile of a Brazilian University Population

**DOI:** 10.3390/life13091925

**Published:** 2023-09-16

**Authors:** Marina dos Santos Barreto, Beatriz Soares da Silva, Ronaldy Santana Santos, Deise Maria Rego Rodrigues Silva, Eloia Emanuelly Dias Silva, Pedro Henrique Macedo Moura, Jessiane Bispo de Souza, Lucas Alves da Mota Santana, Dennyson Leandro M. Fonseca, Igor Salerno Filgueiras, Adriana Gibara Guimarães, Otavio Cabral-Marques, Lena F. Schimke, Lysandro Pinto Borges

**Affiliations:** 1Department of Pharmacy, Federal University of Sergipe, São Cristóvão 49100-000, SE, Brazil; sbarretomarina@outlook.com (M.d.S.B.); biaisas@hotmail.com (B.S.d.S.); ronaldyss19@gmail.com (R.S.S.); deisemaria588@gmail.com (D.M.R.R.S.); eloiaemanuelly@gmail.com (E.E.D.S.); phmm694@gmail.com (P.H.M.M.); jeisse.nik@hotmail.com (J.B.d.S.); adrianagibara@academico.ufs.br (A.G.G.); 2Graduate Program in Dentistry, Federal University of Sergipe, São Cristóvão 49100-000, SE, Brazil; lucassantana.pat@gmail.com; 3Interunit Postgraduate Program on Bioinformatics, Institute of Mathematics and Statistics (IME), University of São Paulo (USP), São Paulo 05508-090, SP, Brazil; dennyson@usp.br; 4Department of Immunology, Institute of Biomedical Sciences, University of São Paulo, São Paulo 05508-000, SP, Brazil; igor.filgueiras@usp.br; 5Department of Medicine, Division of Molecular Medicine, School of Medicine, University of São Paulo, São Paulo 01246-903, SP, Brazil; 6Department of Pharmacy and Postgraduate Program of Health and Science, Federal University of Rio Grande do Norte, Natal 59012-570, RN, Brazil; 7Laboratory of Medical Investigation 29, University of São Paulo School of Medicine, São Paulo 01246-903, SP, Brazil; 8Department of Clinical and Toxicological Analyses, School of Pharmaceutical Sciences, University of São Paulo, São Paulo 05508-000, SP, Brazil; 9Network of Immunity in Infection, Malignancy, Autoimmunity (NIIMA), Universal Scientific Education and Research Network (USERN), São Paulo 05508-000, SP, Brazil

**Keywords:** COVID-19, antibodies, universities, academic population, vaccines, Brazil

## Abstract

Background: COVID-19 led to the suspension academic activities worldwide, affecting millions of students and staff. Methods: In this study, we evaluated the presence of IgM and IgG anti-SARS-CoV-2 antibodies in an academic population during the return to classes after a one-year suspension. The study took place over five months at a Brazilian university and included 942 participants. Results: We found that most participants had reactive IgG and non-reactive IgM. All received at least one dose, and 940 received two or more doses, of different COVID-19 vaccines. We obtained a higher average of memory antibodies (IgG) in participants who received the CoronaVac/ChAdOx1 combination. IgG was consistently distributed for each vaccine group, but individuals who completed the vaccination schedule had higher levels. There were no differences between antibodies and gender, presence of symptoms, and previous COVID-19 infection, but older participants (>53 years) and contacts of infected individuals had higher IgM levels. Conclusion: This study makes significant contributions to the assessment of antibodies in the academic environment, allowing us to infer that most participants had memory immunity and low indications of recent infection when returning to face-to-face classes, as well as demonstrating the need to monitor immunity and update vaccinations.

## 1. Introduction

The coronavirus disease (COVID)-19 pandemic spread rapidly worldwide after its discovery in December 2019 [1]. The World Health Organization (WHO) recommended several preventive measures since no effective treatment was approved for severe acute respiratory syndrome coronavirus 2 (SARS-CoV-2) infection [2]. Among them were hygiene of hands and environmental surfaces, social distancing, and using masks to minimize viral spread [3,4]. In addition, several countries [5], including Brazil, suspended in-person classes in schools and universities in March 2020 [6] when the government declared a Public Health Emergency of National Importance (ESPIN) [7] and reported 291 cumulative cases and the first death confirmed by the Brazilian Ministry of Health [8]. University students generally do not present the severe form of the disease due to their young age and low rate of comorbidities. However, they are highly socially active, representing a critical population spreading the SARS-CoV-2 to older individuals in their residences or other environments [9,10].

During the COVID-19 pandemic, more than 168 million students were affected by the suspension of classes worldwide, and after eighteen months of the pandemic, this number still exceeded 70 million [11]. The withdrawal of students from schools and universities negatively impacted this population’s quality of life and learning development [12,13,14,15,16]. Thus, one of the initiatives that made a safe return of academic activities possible in the first half of 2022 [6] was the development of vaccines against SARS-CoV-2 [17,18,19]. In Brazil, vaccines from Pfizer/BioNTech (BNT162b2), Oxford/AstraZeneca (ChAdOx1), Janssen-Cilag (Ad26.COV2.S), and Sinovac (CoronaVac) have been approved by the Brazilian Health Regulatory Agency (ANVISA) and are available to the population [8]. Vaccines produced with mRNA technology, such as BNT162b2, make the mRNA available for processing and antigen presentation [20]. ChAdOx1 and Ad26.COV2.S vaccines use a non-recombinant viral vector, which does not have the gene responsible for the disease but can enter the host cells by causing the immune system to recognize the remaining viral proteins [21]. Another technology is the chemically or physically inactivated virus, used by CoronaVac, which promotes the recognition of various virus proteins by the immune system without viral infection [22]. Notably, all these immunizers activate the immune response and stimulate antibody production against SARS-CoV-2 from the presentation of the virus antigen [19,23].

Assessing the seroprevalence of anti-SARS-CoV-2 antibodies, either induced by natural infection or by vaccination, during the return to in-person classes is essential for evaluating disease epidemiology and vaccination effectiveness [3,24,25,26,27]. Therefore, here we assessed the serum prevalence of anti-SARS-CoV-2 antibodies in 942 individuals enrolled at the Federal University of Sergipe (UFS) in São Cristóvão, Brazil, which registered more than 34,000 students, employees, and teachers at the time of the university reopening in January 2022 [28]. The SARS-CoV-2 Omicron variant was prevalent at this moment, and was responsible for Brazil’s third wave of COVID-19 [29].

## 2. Materials and Methods

### 2.1. Study Population and Sample Collection

The study was approved by the Research Ethics Committee (CAAE 31018520.0.0000.5546). The inclusion criteria selected study participants without specific underlying diseases, no comorbidities, and who were connected to the Federal University of Sergipe as a student, faculty, or staff member when signing the informed consent. The study included 942 (490 female and 452 male) volunteers between January and May 2022. The Omicron variant of SARS-CoV-2 was the predominant variant in this period [29]. The age of the study participants ranged from 18 to 73 years, with a mean of 30.27 ± 11.38 (95% CI 29.5–31.0). Figure 1 summarizes the workflow of this study. Briefly, a basic questionnaire adapted from the notification forms for COVID-19 of the Brazilian Ministry of Health [8] was applied to obtain gender, date of birth, symptoms of COVID-19 in the last 15 days, recent contact with individuals with a positive COVID-19 diagnosis, and vaccination status. Then, blood samples using gel separator tubes were obtained to evaluate the serum levels of the anti-SARS-CoV-2 antibodies (described below).

### 2.2. Laboratory Analysis

To assess serum levels of anti-SARS-CoV-2 antibodies (Ab), we used the immunofluorescence assays applying the ichroma™ COVID-19 Ab test following the manufacturer’s recommendation, which uses a sandwich immunodetection methodology (https://www.boditech.co.kr/en/support/id/226, accessed on 9 November 2021) [30]. This method has a sensitivity of 95.8% and a specificity greater than 97%, as reported by the manufacturer [31]. This approach allows us to determine the immunoglobulin (Ig)G and IgM levels against SARS-CoV-2. The results obtained were accessible for each participant through a QRcode^®^.

### 2.3. Statistical Analysis and Data Visualization

We used IBM^®^ SPSS^®^ Statistics software (version 26.0 for Windows) [32] for data analysis. The Mann–Whitney and Kruskal–Wallis independent sample tests were used to differentiate antibody distribution between two or more groups. Percentages were calculated. The device calculates the results of antibody levels, expressing them in an auxiliary value, which is displayed as a cut-off index (COI) for IgM and IgG. Values equal to or greater than 1.1 are classified as reactive, and values less than 1.1 as non-reactive. According to the manufacturer’s package insert, values between 0.9 and 1.1 are considered indeterminate. A *p*-value of <0.05 was considered statistically significant. Spearman’s Correlation test was performed for correlation of quantitative variables, with “*” = *p* < 0.05, “**” = *p* < 0.01. Graphs were visualized using jamovi^®^ (version 2.3.28 for Windows) [33] to analyze the distribution of IgG antibodies in each vaccine group and GraphPad Prism^®^ (version 9.5.1 (733) Windows) [34] to investigate the dispersion of IgM and IgG antibodies in individuals who have or have not had contact with someone infected, symptomatic and asymptomatic, and in those who had previous COVID-19 and those who never had. The online web tool Circos^®^ plot (http://circos.ca/, accessed on 17 June 2022) [35] was used to correlate the total number of participants with the number of individuals for each vaccine group.

## 3. Results

We obtained an average mean of 23.84 ± 7.64 COI (95% CI 23.4–24.3; *p* < 0.01) for the IgG antibody and 0.74 ± 1.40 COI (95% CI 0.65–0.83, *p* < 0.01) for IgM. Seven hundred fifty-seven individuals (80.4% of the total number of participants) presented results below 1.1 COI for IgM, and only six (0.6%) showed non-reactive IgG levels (below 1.1 COI). In contrast, 99.4% showed reactive IgG levels. Three (0.23%) individuals had non-reactive results for both types of antibodies (Appendix A). This result indicates that most of the study population presents reactive results for anti-SARS-CoV-2 IgG antibodies. However, only a small number of participants expressed reactive results for IgM.

Evaluating the kind of vaccination received by each individual, we obtained ten different vaccine combinations of four different vaccination types (BNT162b2, ChAdOx1, Ad26.COV2.S, and CoronaVac) offered in the Brazilian territory (Appendix A). As shown in Figure 2, the five groups of vaccines received most by the participants were BNT162b2 (A) (*n* = 400), BNT162b2/ChAdOx1 (AB) (*n* = 226), ChAdOx1 (B) (*n* = 116), CoronaVac/BNT162b2 (BC) (*n* = 78), and BNT162b2/Ad26.COV2.S (AD) (*n* = 49). Combinations with the Ad26.COV2.S vaccine was less present in this study group, probably because this vaccine was the least used in the Brazilian population [8] (Figure 2).

We performed the independent samples Kruskal–Wallis test to assess whether vaccine groups affect the distribution of participants’ IgG antibodies. We determined that the distribution of IgG antibodies is different (*p* = 0.004) across the vaccination groups. Figure 3a shows the distribution of IgG levels in each vaccine group. A higher mean was obtained for the ChAdOx1/CoronaVac (BC) combination (*m* = 27.43 ± 12.19), followed by BNT162b2/CoronaVac (AC) (*m* = 25.37 ± 7.98). BNT162b2 (A) presented a higher average (*m* = 24.66 ± 6.41) in the homologous vaccine groups, followed by Ad26.COV2.S (D) (*m* = 24.16 ± 9.13), CoronaVac (C) (*m* = 23.76 ± 12.01), and ChAdOx1 (B) (*m* = 22.71 ± 8.87) vaccine. The other combinations reached a lower average, as shown in Figure 3b and Appendix A. The BNT162b2 (A) vaccine showed a concentrated result at higher IgG levels. In contrast, other vaccines, such as ChAdOx1 (B), CoronaVac (C), and the combination of the two, showed a more expanded distribution for the antibody, including shallow levels (Figure 3b). It can also be seen that the same group of vaccines generated varied individual responses, contributing to the vast difference in antibody level distribution.

All study participants had used at least one dose of any of the four COVID-19 vaccines. Figure 4 shows the IgM and IgG antibodies distribution for each vaccine dose. Only two participants had not updated their vaccination schedule, leaving them with only one dose administered. The remaining participants received either up to the second vaccination dose (completed the vaccination schedule; two doses group) or had updated the vaccination schedule for the booster dose (third dose of the vaccine, reinforcing the vaccination schedule; booster dose group). The average antibody level for the one-dose individuals (*n* = 2) was 1.00 ± 0.42 for IgM and 17.3 ± 3.54 for IgG. The two-dose group (*n* = 429) and the booster dose group (*n* = 511) showed an average antibody level of 0.59 ± 1.10 and 0.74 ± 1.40 for IgM and 24.41 ± 8.17 and 23.39 ± 7.15 for IgG, respectively. Using the Kruskal–Wallis test of independent samples, we evaluated the distribution of IgM and IgG per different vaccine doses applied to the participants. We discovered that the distribution of IgM was the same (*p* = 0.092), and that IgG antibody levels were distributed differently according to the number of doses (*p* = 0.031). However, the significant difference in sample size between the one-dose group and the two- or booster-dose group may impact this result. Despite this, Figure 4 shows that the distribution of both antibodies is similar in individuals who received two or three vaccine doses. Spearman’s correlation indicates that the number of doses applied increases with age (*r* = 0.276 **, *p* < 0.01). In this regard, it is worth mentioning that the vaccination schedule in Brazil occurred by age group, with older people having priority over younger people [8].

The Kruskal–Wallis test was performed to evaluate the difference in IgG antibody distribution during the months of the study. The analysis points out that the IgG antibodies were distributed differently among the five consecutive months of the study (*p* = 0.001), decreasing their levels progressively during the five months of the study (January to May) (Table 1). It is essential to note that the face-to-face classes returned in January, and in May, the UFS entered the vacation period. This may explain the low adherence to serological testing in this month compared to the other months of the study. Also, it is worth mentioning that the first booster dose was released in November 2021 for the public over 18 years of age and without comorbidity [36], which could have impacted higher IgG levels measured at the beginning of our study in 2022.

Regarding the distribution of antibodies according to sex, the Mann–Whitney test indicates that IgM is distributed differently for females and males (*p* = 0.001), while IgG is equally distributed by sex (*p* = 0.839). However, this difference concerning the mean is not significant, as the total mean IgM levels for females (*m* = 0.74 ± 1.24) and males (*m* = 0.73 ± 1.55) are very similar, as is IgG (*m* = 23.93 ± 7.48 for females and *m* = 23.74 ± 7.82 for males). Table 1 shows the distribution of IgM and IgG antibodies by sex in each month of the study, with similar values between these antibody averages. Spearman’s correlation showed significant correlations between age and IgM (*r* = 0.149 ** and *p* < 0.01) and IgG (*r* = −0.116 ** and *p* > 0.01) (Appendix A), whereas younger individuals tended to have higher titers for IgG. Table 2 shows the averages of IgM and IgG antibodies in age groups, indicating the increase in IgM in older age groups and the increase in IgG in younger groups.

When questioned about flu symptoms in the last 15 days, 182 (19.32%) participants claimed to be symptomatic, and 760 (80.68%) were asymptomatic. Applying the Mann–Whitney test, IgM and IgG were equally distributed in the asymptomatic group compared to the symptomatic group (*p* = 0.057 and *p* = 0.059, respectively) (Figure 5a). The mean IgM for the symptomatic group is 0.95 ± 1.73, and for the asymptomatic group, 0.69 ± 1.30. The mean IgG for the symptomatic groups is 23.00 ± 8.54 and for the asymptomatic group, 24.04 ± 7.40. However, we must emphasize that we did not perform additional diagnostic tests for the presence of the SARS-CoV-2 virus by RT-PCR, so being asymptomatic or symptomatic should not suggest the presence of active infection. In addition, 197 participants reported they had recent contact with someone diagnosed positive for COVID-19, and the remaining 745 had no contact with someone infected. The mean IgM was statistically similar for those individuals with contact with someone positive for COVID-19 (0.80 ± 1.39) and those individuals without contact (0.72 ± 1.40), as was the mean for IgG in these respective groups (24.44 ± 8.58 and 23.68 ± 7.37, respectively. The Mann–Whitney test indicated that there is only a difference in IgM antibody distribution (*p* = 0.022) but not in IgG distribution (*p* = 0.371) regarding contact with COVID-19-positive individuals (Figure 5b).

Regarding the history of diagnosis for SARS-CoV-2, more than half of the participants (*n* = 522) had never been positive for COVID-19, while the remaining (*n* = 420) had already been diagnosed with COVID-19 at least once. The antibodies IgM (*p* = 0.957) and IgG (*p* = 0.110) were equally distributed between the group with positive history of COVID-19 (IgM mean of 0.80 ± 1.54, and IgG mean of 24.29 ± 7.85) and the group that had never been infected (IgM mean of 0.66 ± 1.20, and IgG mean of 23.48 ± 7.46) (Figure 5c) indicating that the antibody levels measured were not influenced by the combination of natural with vaccine-induced immunity or vaccine-induced immunity alone.

## 4. Discussion

Brazil was one of the countries most affected by the COVID-19 pandemic in the number of deaths and cases [37]. To contain the impact of the virus, one of the measures adopted was suspending face-to-face classes and moving to the remote model. In this scenario, millions of students carried out their academic activities from home until this group obtained vaccines against COVID-19 and face-to-face classes could be resumed more than a year after the suspension [6]. Seeking to evaluate the development of the immune response of academics through antibodies, we assessed the prevalence of IgM and IgG anti-SARS-CoV-2 antibodies in an academic population in the first five months of returning to face-to-face classes. Our results suggest that, in general, study participants did not have an acute or recent infection due to the low prevalence of reactive IgM antibodies. However, the high presence of memory antibodies (IgG) against SARS-CoV-2 shows that they had contact with the virus or parts of the virus, either by asymptomatic/mild natural infection or by vaccination. In a previous study by our group conducted one month after the reopening of schools in Sergipe and before vaccination campaigns had started, we reported high virus circulation with few IgG-reactive individuals [38]. In contrast, in this study, we observed a low indication of recent infection, and most participants had already developed memory antibodies, bringing the importance of vaccination to this result.

All vaccines offered in the Brazilian territory and evaluated in this study proved solidly effective in producing memory antibodies. The vaccine combination CoronaVac/ChAdOx1 (group BC in Figure 3) showed the highest mean for IgG. However, 63.6% (7/11) of the individuals who used this combination also reported natural infection, which means that the higher IgG production may be linked to hybrid immunity. This type of immunity results from the combination of infection-induced and vaccine-induced immunity and is associated with increased antibody production [39]. Nevertheless, our study did not generally identify altered seroprevalence concerning vaccine-induced or hybrid immunity or the type of vaccine combination applied. In addition, we did not evaluate the time interval between natural infection, vaccination, and testing time for this study, which may influence the level and prevalence of antibodies subject to natural decay [40]. In addition, Brazil was a country marked by underreporting and under-testing COVID-19 cases due to limited capacities, which may impact this result.

The state Sergipe, where the study was conducted, released the booster dose of the COVID-19 vaccine in the second half of November 2021 for the adult public [36]. When we evaluated the number of vaccine doses and antibody levels, we found that individuals who had only one dose of the vaccine had lower mean IgG levels than those with two doses or a booster dose (third dose), but this may be affected by the significant difference in sample size, where the distribution of participants with one dose is disproportionate to those with two and three doses. Our results also show a trend of progressive decrease in IgG antibody levels over the months, which has already been reported in some studies with vaccinated individuals [40,41]. These findings reinforce the importance of updating the vaccination schedule to maintain antibody levels.

We found no differences in the IgG antibody levels between women and men, corroborating studies that found no differences between sex and IgG seroprevalence [42,43,44,45,46,47]. Despite showing a different distribution between the two groups, the mean IgM for males and females is statistically similar. Not all vaccinated individuals seroconverted to IgG, as found in other studies [48,49], and this may be linked to several extrinsic inter-individual and intrinsic (e.g., genetic) factors interfering with the development of the immune response [50].

Although some studies have found no relationship between age and seropositivity [42,43,44,47,51], other studies corroborate our findings, suggesting that IgG is higher in younger people [46,52,53,54,55,56]. The higher IgG levels in young people may be caused by hybrid immunity, as they are more socially active and at higher risk of infection, which often remains asymptomatic [57]. Therefore, when vaccinated, they may generate a hybrid immune response, which correlates with higher IgG levels [39]. Regarding the correlation between higher levels of IgM and older individuals, a study in a population of blood donors found the same result, attributing it to possible false positives and preferential seroconversion to IgM [53].

This study has some limitations. The fact that not all samples were evaluated on the same date but during a period of five months may cause slight experimental variation in antibody results, as well as behavioral and sociocultural differences in participants. In addition, responses regarding the presence of symptoms, contact with someone infected with SARS-CoV-2, and previous COVID-19 infection were recorded according to the participant’s statement and are, therefore, subject to recall bias. In addition, the experimental workflow of this study did not include diagnostic tests, such as RT-PCR, or neutralizing antibodies and SARS-CoV-2-specific T cells, which would complement our results and further evaluate the immune response to SARS-CoV-2.

In summary, we demonstrated that the academic population showed a significant memory antibody response after vaccination and during the resumption of face-to-face classes, and we did not detect a high prevalence of antibodies signaling active/recent infection in participants. The association of anti-SARS-CoV-2 IgG antibodies with vaccination doses and combination groups did not highlight a more prominent immune response for a specific vaccination. Still, it showed a tendency for higher IgG levels in participants who received more than one vaccination dose. Therefore, besides offering a parameter of seroprevalence in Brazilian universities post-vaccination and post-reopening of universities, our study suggests a possible necessity of vaccine application to establish a robust anti-SARS-CoV-2 immune response. However, it is essential to remember that further research is needed to fully understand the immune response and the efficacy of vaccination in protecting against aggravation and death caused by COVID-19.

## Figures and Tables

**Figure 1 life-13-01925-f001:**
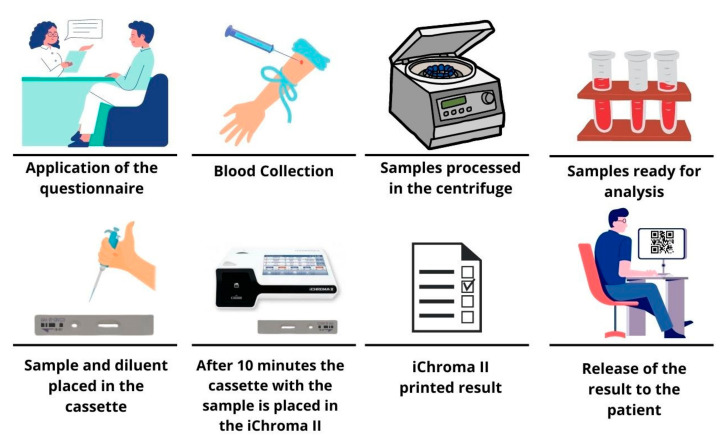
Study workflow showing steps taken to collect serological samples, analyze the IgM and IgG antibody levels, and release results for participants. Initially, we collected demographic data of the participants from a questionnaire. Subsequently, participants were directed to blood puncture. The collected samples were sent to centrifugation and further processed for laboratory analysis. Following sample preparation and insertion into the cassette, samples were analyzed in the ichroma™ COVID-19 device, which performed the reading and expression of the result obtained for the COI value for IgM and IgG antibodies. The results were transmitted to the patient online. Ig: immunoglobulin, COI: Cut-off index.

**Figure 2 life-13-01925-f002:**
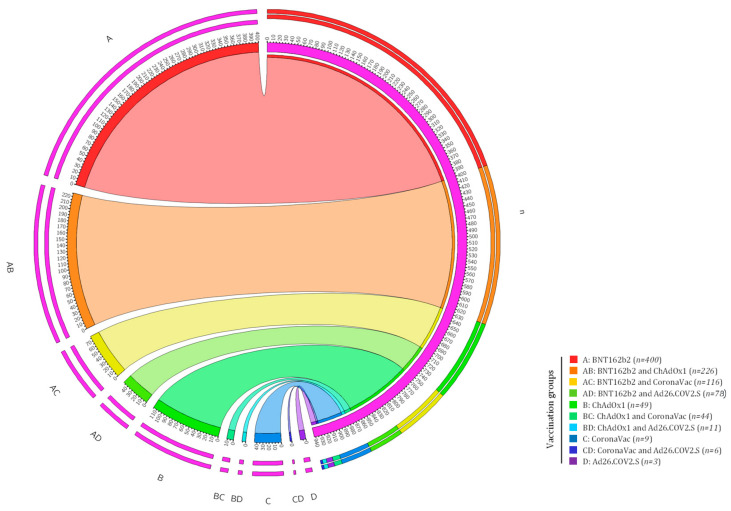
Distribution of the number of participants according to vaccine or vaccination combination group. The graph shows the total number of individuals (*n* = 942) and the distribution according to each vaccine and vaccine combinations applied to the study population. Aside from the Circos^®,^ the figure legend denotes the vaccines and their respective combinations by letters and different ribbon colors.

**Figure 3 life-13-01925-f003:**
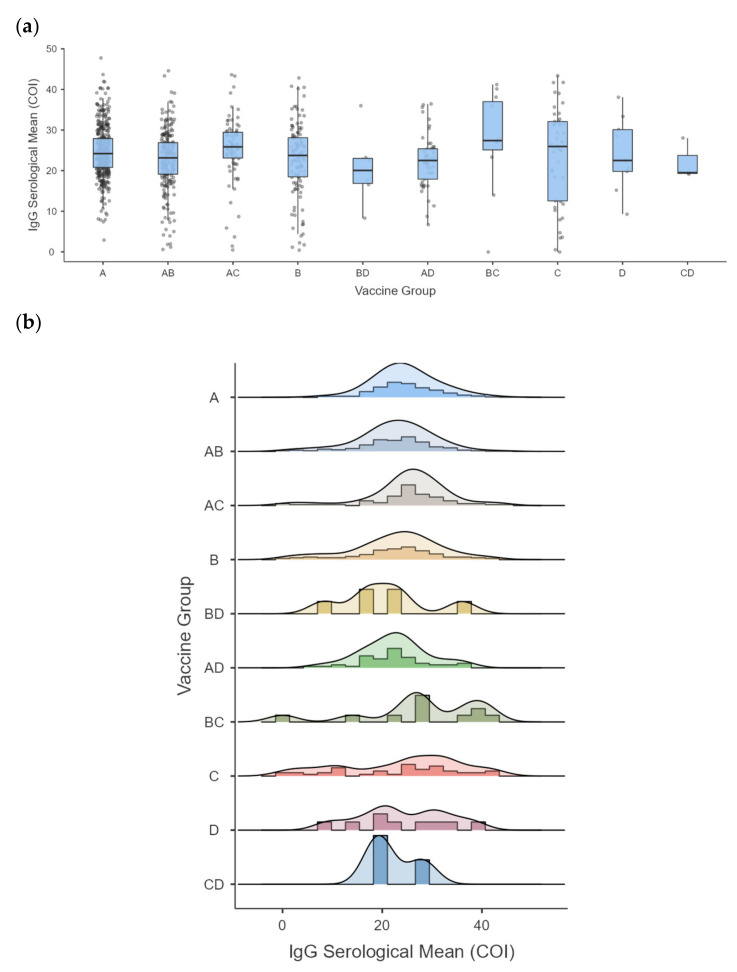
Distribution and means of anti-SARS-CoV-2 IgG antibodies in the different vaccine groups. (**a**) Boxplots show the distribution of IgG values for each vaccine group. Blue boxes indicate the normal distribution of the antibody, with the mean (*m*) flagged in the middle of each box. The black dots represent the outliers, considered abnormal values that deviate from the normal distribution. (**b**) The histograms indicate the antibody level distribution (wave line) and antibody level concentration peaks (colored bars) in each vaccine group. The larger bars indicate that more individuals present the indicated mean value (concentration peak), while smaller bars correspond to fewer individuals with this mean antibody level. Ig: immunoglobulin. The vaccine groups are indicated by letters and different colors, with A = BNT162b2 (*m* = 24.66 ± 6.41), B = ChadOx1 (*m* = 22.71 ± 8.87), C = CoronaVac (*m* = 23.76 ± 12.01) D = Ad26.COV2.S (*m* = 24.16 ± 9.13), AB = BNT162b2 and ChadOx1 (*m* = 22.68 ± 7. 44), AC = BNT162b2 and CoronaVac (*m* = 25.37 ± 7.98), AD = BNT162b2 and Ad26.COV2.S (*m* = 22.40 ± 6.63), BC = ChadOx1 and CoronaVac (*m* = 27.43 ± 12.19), CD = CoronaVac and Ad26.COV2. S (*m* = 22.20 ± 5.03), and BD = Ad26.COV2.S and ChadOx1 (*m* = 20.70 ± 9.21).

**Figure 4 life-13-01925-f004:**
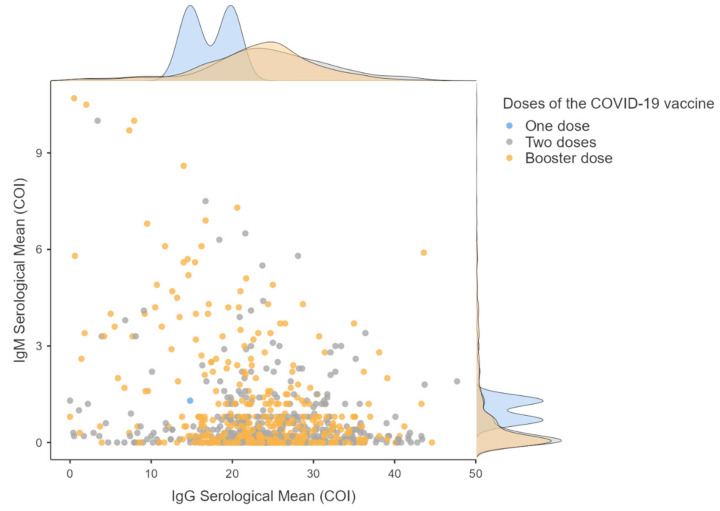
Distribution of IgM and IgG anti-SARS-CoV-2 antibodies in different COVID-19 vaccine dose groups. The scatter plot shows the distribution of IgM and IgG for individuals who used only one vaccine dose (*n* = 2), two doses (*n* = 429), and the booster dose (*n* = 511). The densities, displayed at the top and right side, help visualize the antibodies’ distribution in the three groups analyzed. Ig: immunoglobulin.

**Figure 5 life-13-01925-f005:**
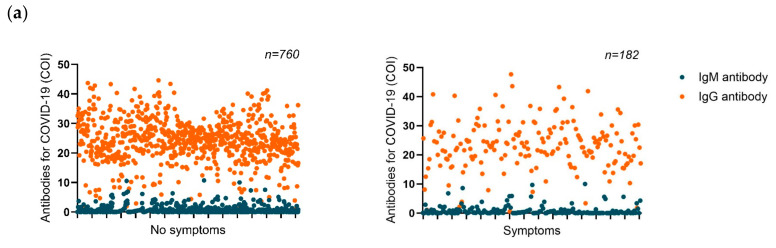
Distribution of anti-SARS-CoV-2 IgM and IgG antibodies among individuals with COVID-19 symptoms, recent contact, or previous natural SARS-CoV-2 infection. (**a**) The scatter plot shows IgM and IgG antibody levels (COI) distribution between symptomatic and asymptomatic individuals for COVID-19 15 days before sample acquisition; (**b**) Distribution of IgM and IgG antibody levels in individuals with recent contact with someone diagnosed with COVID-19 positive and those without contact with COVID-19 individuals; (**c**) IgM and IgG antibody level distribution in individuals who had ever been diagnosed with COVID-19 and those who reported having never been infected with SARS-CoV-2. Ig: immunoglobulin.

**Table 1 life-13-01925-t001:** Distribution of antibody serological mean for anti-SARS-CoV-2 IgM and IgG antibodies during the five months of data acquisition of this study (January to May 2022). The antibody means are divided by sex groups and month of sample acquisition, showing a slight variation in low IgM levels throughout the study but high IgG levels at the beginning with a tendency to decrease as the months of the study pass. Ig: immunoglobulin, CI: Confidence interval.

Antibodies Serological Mean (COI)				
Month (*n*)	Sex (*n*)	IgM	CI 95%	*p*
January (29)	Female (14)	0.40 ± 0.62	0.04–0.76	0.03
	Male (15)	0.69 ± 1.25	0.00–1.38	0.049
February (448)	Female (232)	0.89 ± 1.38	0.71–1.07	0.001
	Male (216)	0.75 ± 1.51	0.55–0.96	0.001
March (279)	Female (151)	0.57 ± 0.99	0.41–0.73	0.001
	Male (128)	0.84 ± 1.84	0.52–1.16	0.001
April (161)	Female (86)	0.71 ± 1.30	0.44–1.00	0.001
	Male (75)	0.54 ± 1.26	0.25–0.83	0.001
May (25)	Female (7)	0.53 ± 0.55	0.02–1.04	0.045
	Male (18)	0.60 ± 1.28	−0.04–1.24	0.064
	Sex (*n*)	IgG	CI 95%	*p*
January (29)	Female (14)	27.27 ± 4.17	24.86–29.68	0.001
	Male (15)	26.21 ± 7.97	21.80–30.63	0.001
February (448)	Female (232)	24.0 ± 8.68	22.92–25.17	0.001
	Male (216)	24.64 ± 8.25	26.53–25.74	0.001
March (279)	Female (151)	24.07 ± 5.22	23.23–24.91	0.001
	Male (128)	23.04 ± 6.99	21.82–24.27	0.001
April (161)	Female (86)	23.10 ± 7.68	21.46–24.75	0.001
	Male (75)	22.71 ± 6.95	21.11–24.31	0.001
May (25)	Female (7)	20.81 ± 7.45	13.92–27.71	0.001
	Male (18)	20.21 ± 9.93	15.27–25.15	0.001

**Table 2 life-13-01925-t002:** Distribution of means for IgM and IgG antibodies according to different age groups. The table shows the variation of the means of the antibody levels for each age group, with IgM levels tending to increase with age and IgG levels higher in younger groups. Ig: immunoglobulin, CI: Confidence interval.

Age Group	*n*	Antibody	Mean (SD)	95% CI	*p*
18–23 Years	373	IgM	0.54 ± 1.13	0.43–0.66	0.001
		IgG	24.51 ± 7.69	23.73–25.30	0.001
24–29 Years	216	IgM	0.54 ± 0.95	0.42–0.67	0.001
		IgG	24.51 ± 6.78	23.60–25.42	0.001
30–35 Years	86	IgM	0.83 ± 1.12	0.59–1.08	0.001
		IgG	22.50 ± 7.75	20.84–24.16	0.001
36–41 Years	88	IgM	1.26 ± 1.92	0.85–1.67	0.001
		IgG	23.15 ± 8.00	21.46–24.85	0.001
42–47 Years	70	IgM	0.68 ± 1.11	0.42–0.94	0.001
		IgG	23.69 ± 7.18	21.98–25.41	0.001
48–53 Years	54	IgM	1.17 ± 1.81	0.67–1.66	0.001
		IgG	23.49 ± 7.42	21.46–25.51	0.001
>53 Years	55	IgM	1.50 ± 2.72	0.77–2.24	0.001
		IgG	20.41 ± 9.45	17.85–22.96	0.001

## Data Availability

If you are interested in further data on the results, please contact the corresponding author.

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
