# Peer review of "COVID-19 Vaccination and Serological Profile of a Brazilian University Population"

_life, 2023, doi:10.3390/life13091925_

Round 1

Reviewer 1 Report

In this manuscript, Marina dos Santos Barreto et al., examined the serological profile (IgG and IgM against SARS-CoV-2). They collected samples in large group in Brazil and found that most participants who received CoronaVac/ChAdOx1 had high reactive IgG. Although there were statistically no differences between Ab and gender, older participants and contacts of infected individuals had higher IgM. Overall, the manuscript was well organized and written. I have a few comments for this manuscript.

1.     Do the authors know if the participants had specific underlying diseases? If so, did the authors see any correlations the specific underlying diseases and anti-SARS-CoV-2 antibodies?

2.     Line 92-93: Which variants the participants were infected during sample correction (between Jan and May 2022 in Brazil)? Please state this in the Materials and methods.

3.     Line number “278” is located in the center of the Figure 5. To avoid the confusion, please fix it.

Minor editing of English language required

Reviewer 2 Report

Here the authors analyzed 942 samples collected from the population in the university and compared SARS2 IgG and IgM. They compared antibodies responses in different vaccination, different dose, vary infection condition.

This is a well-designed study to show a original data that expand the understanding in sars2 vaccination and antibody responses. I thought the study was rigorous and only one question addressed below.

Line142, Line 169, can’t find Supplementary Tables. The authors need doublecheck it.
